# Citron Watermelon Potential to Improve Crop Diversification and Reduce Negative Impacts of Climate Change

**Takudzwa Mandizvo [1],\***, **Alfred Oduor Odindo [1]** and **Jacob Mashilo [2]**

[1] Crop Science, School of Agricultural Earth and Environmental Sciences, University of KwaZulu-Natal, Pietermaritzburg 3201, South Africa; odindoa@ukzn.ac.za

[2] Limpopo Department of Agriculture and Rural Development Towoomba Research Station, Private Bag X1615, Bela-Bela 0480, South Africa; jacobmashilo@yahoo.com

\* Correspondence: takudzwamandizvo@gmail.com; Tel.: +27-629810895

**Abstract:** Citron watermelon (*Citrullus lanatus* var. *citroides* (L.H. Bailey) Mansf. ex Greb.) is an underexploited and under-researched crop species with the potential to contribute to crop diversification in Sub-Saharan Africa. The species is cultivated in the drier parts of Southern Africa, mainly by smallholder farmers who maintain a wide range of landrace varieties. Understanding the molecular and morpho-physiological basis for drought adaptation in citron watermelon under these dry environments can aid in the identification of suitable traits for drought-tolerance breeding and improve food system resilience among smallholder farmers, thus adding to crop diversification. This paper reviews the literature on drought adaptation of *Citrullus lanatus* spp. (C3 xerophytes), using the systematic review approach. The review discusses the potential role of citron watermelon in adding to crop diversification, alternative food uses, and potential by-products that can be processed from the crop, and it analyzes the role of Sub-Saharan African farmers play as key actors in conserving citron watermelon germplasm and biodiversity. Finally, the review provides a summary of significant findings and identifies critical knowledge gaps for further research.

**Keywords:** abiotic stress; *C. lanatus* var. *citroides*; drought adaptation; food security; underutilized crops

## 1. Introduction

Citron watermelon (*Citrullus lanatus* var. *citroides* (L.H. Bailey) Mansf. ex Greb.) belonging to the *Cucurbitaceae* family originated in Southern Africa [1]. It is a facultative xerophyte following the C3 photosynthetic pathway [2,3]. The citron watermelon plant is a vine creeper with herbaceous stems up to 3 m long. Young stems and leaves are densely woolly, while the older parts become hairless. The leaves are herbaceous, sometimes unlobed, but usually 3-lobed. Both female and male flowers are on the same plant (monoecious) [4]. The fruits are formed in different shapes (subglobose, indehiscent globose, ellipsoid, or oblong) and can be up to 200 mm in diameter. The rind of ripe fruit is hairless and smooth with different colors, which are usually mottled with irregular longitudinal bands [5]. The flesh is firm and white, green-white or yellowish. The seeds are dicotyledonous, and typically red, white, or mottled in seed coat color.

Citron watermelon is mainly produced in Southern Africa [3]. Clustered data FAOSTAT [6] on melons (including cantaloupes) production show that production has declined in Southern Africa from 1990 to 2017 by approximately 44%. The decline in production is attributed to pests and diseases, drought, and poor agronomic practices. Although until the beginning of the 1980s, the cultivation of citron watermelon was specific to countries in Southern Africa, other regions have understood the potential and benefits of the crop. Consequently, both the research and production of citron watermelon have been growing steadily worldwide [4].

The young tender leaves of citron watermelon can be cooked as a green vegetable, while mature fruit flesh is mixed with maize meal to prepare porridge [7]. When eaten fresh, the flesh has a non-bitter and blunt taste and is often used to prepare preserves by adding sugar. The fleshy pulp contains pectins, which are processed to make perfect preserves. Citron watermelon fruit and vines is a valuable livestock feed during drought [8]. The seeds can be dried, roasted, and eaten or ground into flour to make condiments. However, despite being a nutrition source, farmers habitually discard citron watermelon seeds, sparing a few to plant in the next season [9].

Promoting the production and use of under-researched crops such as citron watermelon could offer potential solutions to mitigate climate change's negative impacts (crop failure, hunger, and malnutrition) [10]. This will contribute toward achieving the Sustainable Development Goals (SDGs) of the 2030 agenda, such as SDG 2 (zero hunger) and other interconnected goals such as SDG 1 (no poverty) and SDG 13 (climate change) [11].

The crop can add to crop diversity, boost food security and local economies, strengthen rural development, and promote sustainable land use. The inclusion of measures to encourage crop diversification among smallholder farmers (who focus on growing few crops) is crucial in ensuring a broad food base and balanced nutrition for populations (rural and urban) in developing countries. Therefore, crop diversification beyond over-reliance on a few food crops is vital in achieving food security.

Despite its potential contribution to food security, both national and international research systems have overlooked citron watermelon and other local and indigenous crops because governments and policy-makers do not see their value as food or cash crops. Consequently, governments hardly prioritize resources (funds) to promote their research [10]. The impact of having policy and government support for agricultural research is evident from the support given to the staple food crops (rice, wheat, maize, and beans), which dominate human diets and have had their yields and nutritional values boosted over the years through breeding. For example, these staple food crops have had their genomes mapped to the level of individual base pairs [12]. There is a need to focus on local or indigenous crops that research programs have previously neglected because of their potential to add value and contribute to food security and improved livelihoods for smallholder farmers.

Research on previously neglected indigenous crops is increasingly being recognized and is receiving continental attention in recent times. For example, in 2011, the New Partnership for Africa's Development (NEPAD) committed to lead a consortium of companies, scientific bodies, and government bodies to sequence, assemble, and annotate the genomes of 100 of Africa's important but neglected food crops [13]. This consortium, African Orphan Crops (AOC) (*africanorphancrops.org*), aimed to train African scientists in plant breeding techniques to breed and improve the sequenced under-utilized crops. These will allow African farmers to grow highly nutritious, productive, and robust crops, creating surpluses for the market.

Citron watermelon has received relatively little research attention as one of the neglected crops on the AOC list. Therefore, the current knowledge on citron watermelon's potential to improve crop diversity and contribute to improved livelihoods in smallholder farming systems, and within the context of climate change effects is reviewed. Firstly, factors associated with citron watermelon adaptation to drought at morpho-physiological and molecular levels and how this relates to yield performance are revised with a systematic review approach. Secondly, citron watermelon's potential role in adding to crop diversification in the smallholder farming systems is presented. Thirdly, alternative food uses and potential by-products that can be developed from citron watermelon for small-scale processing and value addition are discussed. Fourth, Sub-Saharan farmers' role as key actors in the conservation of citron watermelon biodiversity is reviewed. Finally, the review summarizes significant findings and identifies critical knowledge gaps for further research.

## 2. Materials and Methods

A systematic review approach was used to map the existing literature supporting the topic's broad research question. The systematic review methodology was based on the framework outlined by Koutsos et al. [14]. The review included the following six steps (Figure 1):

1. Scoping
2. Planning
3. Identification/search
4. Screening
5. Eligibility/assessment and
6. Presentation/interpretation

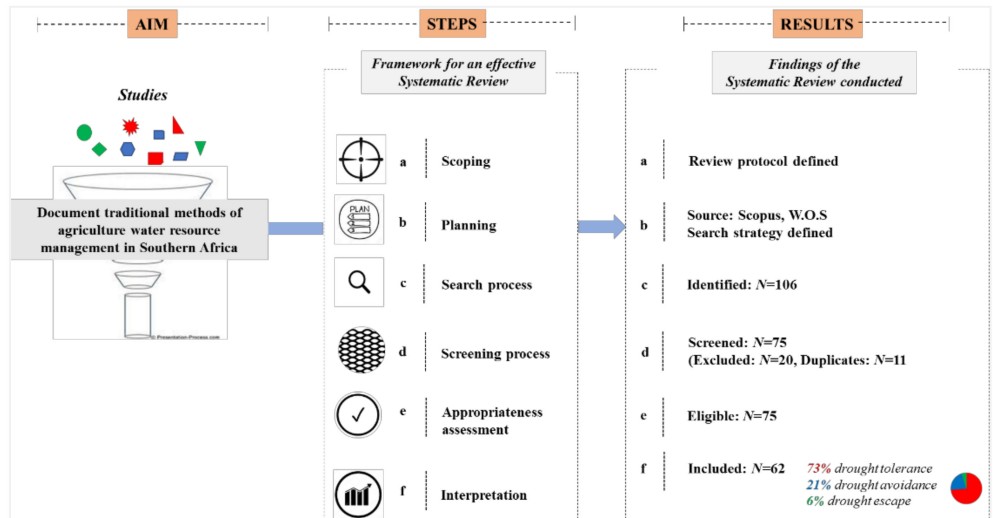

**Figure 1.** The framework used to perform a systematic review for current drought stress tolerance mechanisms in *C. lanatus* spp. (Adopted: Koutsos et al. [14]).

### 2.1. Research Question

This review was guided by the question, "What is the current understanding of drought stress tolerance mechanisms in *Citrullus lanatus* spp.?" Citron watermelon is a C3 crop that is known to be less photo-efficient. It is worthy of understanding how this species deals with the problem of Rubisco having an affinity for oxygen at low $CO_2$ concentrations.

### 2.2. Data Sources and Search Strategy

The search was implemented in five electronic databases:

1. Scopus (www.scopus.com) (accessed on 8 February 2020)
2. Web of Science (www.webofknowledge.com) (accessed on 8 February 2020)
3. Science Direct (www.sciencedirect.com) (accessed on 13 February 2020)
4. Science.gov (www.science.gov) (accessed on 23 February 2020) and
5. Google Scholar (scholar.google.com) (accessed on 10 March 2020)

These databases were selected to be inclusive and cover disciplines in agriculture sciences. Limits on the database search included peer-reviewed literature published from 1 January 1995 to 31 December 2019. The date range limitation was chosen to focus on contemporary literature on drought tolerance mechanisms. The search strategy employed broad search terms (Table 1) to ensure publications were not overlooked.



**Table 1.** Search strategy with Boolean operators for each database to identify peer-reviewed articles examining drought stress tolerance mechanisms in *C. lanatus* spp.

| Database (s) | Primary Term (s) | Expanded Term (s) |
|---|---|---|
| Scopus Web of Science Science Direct Science.gov Google Scholar | Drought stress | "water stress" OR "moisture stress" OR "water deficit" OR "water shortage" OR "water scarcity" AND |
| | *C. lanatus* spp. | "watermelon" OR "citron watermelon" OR "desert watermelon" OR "wild watermelon" OR "melon" OR "muskmelon" AND |
| | Mechanism(s) | "adaptation strategy" OR "avoidance" OR "escape" OR "tolerance" |

### 2.3. Citation Management

Citations were imported into the DistillerSR (Evidence Partners Incorporated, Ottawa, ON, Canada) web-based application, and duplicate citations were removed using the duplicate removal function of DistillerSR. Subsequently, the title and abstract relevance screening and data characterization of complete articles were carried out using DistillerSR.

### 2.4. Relevance Screening and Eligibility Criteria

A two-step screening relevance technique was employed. For the first step of screening, the titles and abstracts of the articles were examined for relevance. Next, all citations considered relevant after the title and abstract screening went through a full-text review. Studies were eligible for inclusion if they were original articles on citron watermelon or *C. lanatus* spp. relevant to drought stress tolerance.

### 2.5. Data Charting

The data collection categories included author, year of publication, drought adaptation strategy, and key results. The data were compiled in a spreadsheet using the DistillerSR report function and subsequently imported into Microsoft Excel 2016.

### 2.6. Summarizing and Reporting

A narrative synthesis approach was used to provide an overview of the existing literature. Firstly, a summary of the study findings was combined, considering the variations that may affect the generalization of drought tolerance mechanisms. Then, study results were organized into categories (drought avoidance, drought tolerance, and drought escape) using thematic analysis techniques [15].

### 3. Results

Overview of Studies Identified

The review resulted in three main themes explaining drought adaptation mechanisms in *C. lanatus* spp. The three main mechanisms are drought avoidance (DA), drought tolerance (DT), and drought escape (DE). Table 2 summarizes 62 mechanisms from 52 articles from a systematic review on drought adaptation mechanisms conducted in the past 25 years (1995–2019). The drought tolerance (DT) mechanism had the highest number of articles (74%), followed by (DA) 21% and (DE) 5% (Figure 2).

In the past 25 years, scientists have investigated the morphology, genetic mechanisms, and molecular mechanisms of drought response to enhance the drought tolerance in *Citrullus lanatus* spp. Drought tolerance mechanisms (45 articles) were more often investigated rather than DA (13 articles) and DE (4 articles). Drought tolerance studies in *Citrullus lanatus* spp. included change in gene expression [16–26] and accumulation of osmolytes (citrulline, glutamate, arginine) in leaves and roots [27–34]. Drought avoidance studies include reduced leaf water loss [35–38], enhanced water uptake [39–42], and accelerated transition from vegetative growth to reproductive growth to avoid complete abortion at the severe drought stress stage [38,43].

**Table 2.** Summary of studies on drought adaptation strategies in *C. lanatus* spp. (drought avoidance (DA), drought tolerance (DT), drought escape (DE)).

| Author | Title | Research summary | DA | DT | DE |
|---|---|---|:---:|:---:|:---:|
| | | | | **Mechanisms** | |
| [44] | Expressed sequence tag-based gene expression analysis under drought stress in wild watermelon | Changes in gene expression in roots within 6 h of water stress. Genes involved in oxidative stress (glutathione peroxidase, glucose-6-phosphate-dehydrogenase, and ascorbate peroxidase) were demonstrated to be regulated by water stress. | | ✓ | |
| [45] | Analysis of drought-induced metallothionein in wild watermelon | Gene (*CLMT2*) of the same homology with type-2 metallothionein contributed to the survival of wild watermelon under severe drought. | | ✓ | |
| [46] | Agrobacterium-mediated transformation system for the drought and excess light stress-tolerant wild watermelon (*Citrullus lanatus*) | Experimental basis for molecular studies of wild watermelon genes helps understand their contribution to stress tolerance in this plant. | | ✓ | |
| [16] | Potent hydroxyl radical-scavenging activity of drought-induced type-2 metallothionein in wild watermelon | Gene (*CLMT2*) of the same homology with type-2 metallothionein contributed to the survival of wild watermelon under severe drought. | | ✓ | |
| [47] | Functional analysis of DRIP-1, a drought-induced polypeptide in wild watermelon | Wild watermelon accumulates high concentrations of citrulline, glutamate, and arginine in its leaves during drought. | | ✓ | |
| [48] | Dynamic changes in the leaf proteome of a C3 xerophyte, *Citrullus lanatus* (wild watermelon), in response to water deficit | Defense response of wild watermelon involves orchestrated regulation of functional proteins, of which HSPs play a pivotal role in the protection of the plant under water deficit. | | ✓ | |
| [49] | Molecular responses of wild watermelon to drought stress | Rapid accumulation of HSPs in stressed melons. | | ✓ | |
| [35] | Potential involvement of drought-induced Ran GTPase CLRan1 in root growth enhancement in a xerophyte wild watermelon | Ran GTPase (CLRan1), expressed in the roots of drought-resistant wild watermelon, functions as a positive factor for maintaining root growth under osmotic stress. | ✓ | ✓ | |
| [50] | Drought mediated physiological and molecular changes in muskmelon (*Cucumis melo* L.) | Increased activity of catalase (CAT), superoxide dismutase (SOD), ascorbate peroxidase (APX), and guaiacol (POD). Under drought stress, muskmelon elevates the abundance of defense proteins and suppresses catabolic proteins | ✓ | ✓ | |
| [17] | Identification, molecular characterization, and expression analysis of RPL24 genes in three Cucurbitaceae family members: cucumber, melon, and watermelon | Ribosomal protein L24 (RPL24) is responsible for stabilizing the peptidyl transferase activity—increased expression of CmRPL24-01 genes in melon leaf tissue at 3 h upon polyethylene glycol treatment. | | ✓ | |
| [51] | Identifying sources of water stress tolerance from wild species of the family *Cucurbitaceae* in vitro culture | Evaluating water stress tolerance capacity using the calus recuperation after dehydration under a laminar flux hot until a loss of 50% of their fresh weight as a basis. | | | ✓ |
| [18] | Genome-wide identification and comparative expression analysis of LEA genes in watermelon and melon genomes | Induction of LEA genes in root and leaf tissues after drought application. | | ✓ | |
| [52] | Foliar application of abscisic acid and sulfonamide compounds induced drought tolerance in watermelon | Sulfacetamide and sulfasalazine improve drought resistance similar to ABA by increasing proline, glycine betaine and malondialdehyde, and ascorbate peroxidase activity. | | ✓ | |
| [36] | The apocarotenoid beta-cyclocitric acid elicits drought tolerance in plants | Volatile compound β-cyclocitral (β-CC) in plant leaves, when converted to β-cyclocitric acid (β-CCA), signals drought tolerance. | ✓ | ✓ | |
| [53] | Assessment of watermelon accessions for salt tolerance using stress tolerance indices | GMP and STI indices indicated that G04, G14, and G21 could be prominent sources to develop drought tolerance. | | | ✓ |
| [43] | Effect of water stress on the carbohydrate metabolism of *Citrullus lanatus* seeds during germination | Stressed seeds lower the rate of $^{14}CO_2$ release from [2-$^{14}$C]acetate, [1-$^{14}$C]glucose, and [6-$^{14}$C]glucose. | ✓ | ✓ | |

**Table 2.** *Cont.*

| Author | Title | Research summary | DA | DT | DE |
|---|---|---|:---:|:---:|:---:|
| [37] | Physiological responses of two contrasting watermelon genotypes exposed to drought and nitric oxide | Drought stress decreased fresh and dry weights of shoots and roots and lengths drought-sensitive genotype KAR. 147 | ✓ | | |
| [54] | Biochemical effects of drought stress on two Turkish watermelon varieties are different and influenced by nitric oxide | MDA (a marker of oxidative damage on lipid membranes) was increased due to drought in watermelon genotypes and NO treatment slightly reduced MDA contents under drought stress. | | ✓ | |
| [19] | Expression analysis of five arabidopsis PDLP5 homologous in watermelon subjected to biotic and abiotic stresses | Plasmodesmata-located protein 5 (PDLP5) controls cell-to-cell communication and defense signaling. | | ✓ | |
| [39] | Establishment of a transgenic hairy root system in wild and domesticated watermelon (*Citrullus lanatus*) for studying root vigour under drought | A powerful tool for the comparative study of the molecular mechanism underlying drought-induced root growth in desert plants. | ✓ | ✓ | |
| [55] | Watermelon (*Citrullus lanatus*) late-embryogenesis abundant group 3 protein, ClLEA3-1, responds to diverse abiotic stresses | A representative group of LEA proteins ClLEA3-1 (Cl017745) can be used as an abiotic stress marker gene in watermelon. | | ✓ | |
| [20] | Abiotic stress and tissue-specific reference genes for quantitative reverse transcription PCR analysis in Korean native watermelons, *Citrullus lanatus* 'Black-King' and 'Speed-Plus-Honey' | Reference genes (ClACT and ClEF1$\alpha$) were expressed in flowers, leaves, tendrils, stem, and roots after drought treatment. | | ✓ | |
| [38] | Preferential decay of the CF1 epsilon subunit induces thylakoid uncoupling in wild watermelon under drought stress | Selective decomposition of epsilon subunit induces uncoupling of thylakoid membranes under drought and contributes to the avoidance of over-acidification in the thylakoid lumen under excess light conditions. | ✓ | | |
| [21] | Expression analysis of beta-glucosidase genes that regulate abscisic acid homeostasis during watermelon (*Citrullus lanatus*) development and under stress conditions | Beta-glucosidase genes regulate ABA content during drought stress. | | ✓ | |
| [40] | Water relations and abscisic-acid levels of watermelon as affected by rooting volume restriction | ABA act as a signal for reduced growth of plants under rooting volume restriction (RVR) conditions. | ✓ | ✓ | |
| [56] | Comparative effects of ethylene inhibitors on Agrobacterium-mediated transformation of drought-tolerant wild watermelon | Controlling ethylene level during co-cultivation and shoot formation using the *cad's-harbour*ing Agrobacterium enhances drought tolerance. | | ✓ | |
| [57] | Proteomic analysis of avoidance and defence mechanisms to drought stress in the root of wild watermelon | Defense response of wild melons involves the orchestrated regulation of functional proteins. | ✓ | ✓ | |
| [58] | Co-expression of cytochrome b$_{561}$ and ascorbate oxidase in leaves of wild watermelon under drought and high light conditions | Levels of cDNA (CLb561A) mRNA and protein were elevated in the leaves during drought. | | ✓ | |
| [59] | Genome-wide expression profiling of leaves and roots of watermelon in response to low nitrogen | Under abiotic stress, leaf tissues are more sensitive compared with root tissues. 9598 genes were differentially expressed, out of which 4533 genes were up-regulated, and 5065 genes were down-regulated. | | ✓ | |
| [41] | Mycorrhizal inoculation affects arbuscular mycorrhizal diversity in watermelon roots but leads to improved colonization and plant response under water stress | Plant inoculation with mycorrhizal fungi was related to the response of plants to water stress conditions by improving WUE. | ✓ | | |

**Table 2.** *Cont.*

| Author | Title | Research summary | DA | DT | DE |
|:---:|:---:|:---:|:---:|:---:|:---:|
| [42] | Ectopic expression of Arabidopsis H$^+$-pyrophosphatase AVP1 enhances drought resistance in bottle gourd (*Lagenaria siceraria* Standl.) | Wild-type plants showed minimal growth while the *AVP*-1 expressing plants resumed rapid growth, displaying longer ramified primary roots. | ✓ | | |
| [22] | Molecular cloning and in silico analysis of DREB-like gene in watermelon | DREB genes were identified from watermelon related to drought-tolerant genes. | | ✓ | |
| [60] | Factors Affecting Germination of Citron melon (*Citrullus lanatus* var. *citroides*) | Citron melon can grow in a wide range of climatic conditions. | ✓ | ✓ | |
| [61] | Vegetable grafting: a toolbox for securing yield stability under multiple stress conditions | Movement of mRNA through the phloem from rootstock to scion regulate plant growth and adaptation to drought stress. | | | ✓ |
| [27] | Phytohormones regulate accumulation of osmolytes under abiotic stress | Osmolytes (proline, glycine-betaine, polyamines, and sugars) are accumulated to safeguard the cellular machinery. Phytohormones (abscisic acid, brassinosteroids, cytokinins, ethylene, jasmonates, and salicylic acid) modulates the accumulation of osmolytes. | | ✓ | |
| [23] | Cloning and expression analysis of the Ccrboh gene encoding respiratory burst oxidase in *Citrullus colocynthis* and grafting onto *Citrullus lanatus* (watermelon) | Drought-responsive gene *Ccrboh* is functionally important during the acclimation of plants to stress, and it is promising for improving the drought tolerance of other cucurbit species. | | ✓ | |
| [62] | Gene expression changes in response to drought stress in *Citrullus colocynthis* | During drought, stress-responsive genes and plant hormones are involved in an extensive cross-talk. | | ✓ | |
| [28] | Biochemical and molecular characterization of glutamate N-acetyltransferase involved in citrulline accumulation in wild watermelon during drought/strong-light stresses | Exogenous acetylene triggers the accumulation of citrulline to maintain the plant membrane structure. | | ✓ | |
| [29] | Regulation of metabolic pathways for the massive accumulation of citrulline during drought/strong light stress in wild watermelon | Rapid accumulation of citrulline in leaves and shoot under drought stress. | | ✓ | |
| [24] | Comparative identification, characterization, and expression analysis of bZIP gene family members in watermelon and melon genomes | Genes (ClabZIP and CmbZIP) were expressed in leaf and root tissues after the drought was imposed. | | ✓ | |
| [25] | Comparative analysis of calcium-dependent protein kinase in Cucurbitaceae and expression studies in watermelon | The study provides insights into the evolutionary history of gene families in Cucurbitaceae and indicates a subset of candidate genes for functional characterizations. | | ✓ | |
| [30] | Glycinebetaine biosynthesis in response to osmotic stress depends on jasmonate signalling in watermelon suspension cells | Osmotic stress-induced glycinebetaine biosynthesis occurs via JA signal transduction and contributes to osmotic stress hardening. | | ✓ | |
| [31] | Genome-wide identification and expression analysis of NF-Y transcription factor families in watermelon (*Citrullus lanatus*) | The study provides a foundation for further functional analysis of NF-Y proteins during watermelon development and responses to drought stress. The results will be valuable for evolutionary analysis of the NF-Y family in *Cucurbitaceae* species. | | ✓ | |
| [63] | Identification and expression analyses of WRKY genes reveal their involvement in growth and abiotic stress response in watermelon (*Citrullus lanatus*) | A total of 63 putative WRKY genes in watermelon were reported to regulate respective target genes. | | ✓ | |
| [32] | Citrulline and DRIP-1 protein (ArgE homologue) in drought tolerance of wild watermelon | Wild watermelon accumulates high concentrations of citrulline, glutamate, and arginine in its leaves during drought. | | ✓ | |

**Table 2.** *Cont.*

| Author | Title | Research summary | DA | DT | DE |
|---|---|---|---|---|---|
| | | | **Mechanisms** | | |
| [33] | Programmed proteome response for drought avoidance/tolerance in the root of a C₃ xerophyte (wild watermelon) under water deficits | Defense response of wild watermelon involves orchestrated regulation of functional proteins, of which HSPs play a pivotal role in the protection of the plant under water deficit. | | ✓ | |
| [64] | Proteomic analysis of drought/strong light stress responses in wild watermelon leaves | DREB genes were identified from watermelon related to drought-tolerant genes. | | ✓ | |
| [65] | Identification of drought-responsible proteins in the root of wild watermelon by proteomic analysis | DREB genes were identified from watermelon related to drought-tolerant genes. | | ✓ | |
| [66] | Regulation of the root development mechanism involved in Ran GTPase of wild watermelon under drought stress | Ran GTPase genes function in watermelon development, as well as in response to abiotic stress and hormones. | | ✓ | |
| [26] | Genome-wide identification and expression analysis of ClLAX, ClPIN, and ClABCB genes families in Citrullus lanatus under various abiotic stresses and grafting | Expression genes (ClLAX, ClPIN, and ClABCB) under drought helps to understand the roles of auxin transporter genes in watermelon adaptation to environmental stresses. | | ✓ | |
| [34] | Antioxidant enzymes activities in leaves and yield analysis of different ecological types watermelon under drought stress | The content of proline and the activity of SOD, POD, and CAT were genotype related. | ✓ | ✓ | ✓ |
| [67] | Identification and characterization of the glutathione peroxidase (GPX) gene family in watermelon and its expression under various abiotic stresses | ClGPX genes function in watermelon development as well as in response to abiotic stress and hormones. | | ✓ | |

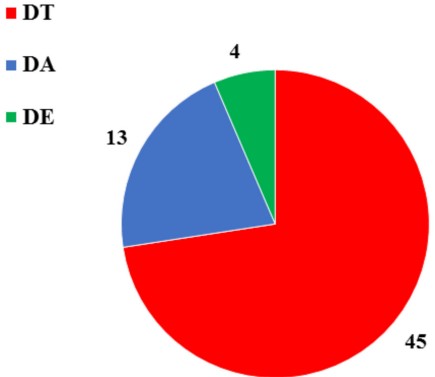

**Figure 2.** Pie chart summarizing 62 drought tolerance mechanisms in *C. lanatus* spp. from 52 articles.

## 4. Discussion

In response to water deficits or drought stress conditions, desert xerophytes have evolved a series of mechanisms at morphological, physiological, and molecular levels to proceed with normal plant function and metabolism. These mechanisms include drought escape (DE) through early completion of a plant life cycle, drought avoidance (DA) through improved water absorbance capacity by the improved root system and shedding of leaves (Figure 3). Drought tolerance (DT) occurs through alteration of the metabolic pathway (for example, increased antioxidant metabolism).

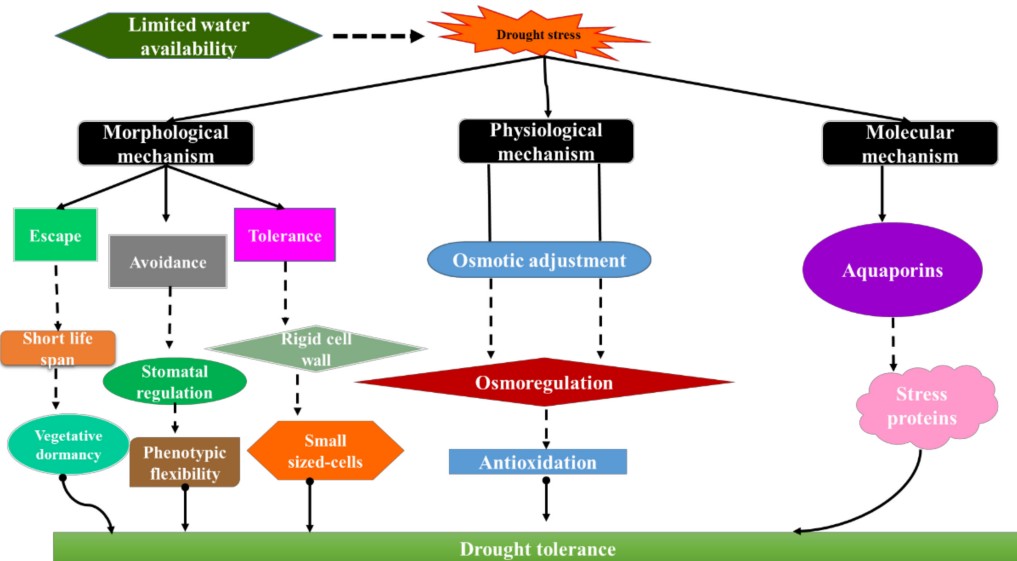

**Figure 3.** Plants adopting morphological, physiological, and molecular behavior under drought stress (Author's drawing).

*Citrullus lanatus* spp. can minimize water loss through transpiration by maintaining basic physiological processes under drought stress conditions, that is, adjusting morphological features (hairy leaf surface and shedding leaves) [68,69]. Primarily, DA is characterized by maintaining high plant water potentials under water stress conditions [70]. *Citrullus. lanatus* spp. has been reported to respond to drought using the following DA mechanisms: (i) reducing water loss by partial closure of stomatal pores, leaf rolling [71], and wax accumulation on the leaf surface [72,73]; (ii) enhancing water uptake ability through a well-developed ramified root system [42], and (iii) accelerating/decelerating the transition from vegetative growth to reproductive growth to avoid complete abortion under severe drought stress [38].

Citron watermelon escapes drought by adjusting its growth period [74]. Plant phenology has a decisive effect on yield under water stress conditions. Early maturity in citron watermelon [35] correlates with root length density to leaf area ratio, which translates to the plant's ability to maintain high leaf water potential under soil moisture stress. Plants that escape drought, such as the desert ephemeral (*Alyssum alyssoides* L.), exhibit early flowering, short plant life cycle, and developmental plasticity [75]. Citron watermelon was reported to escape drought through early flowering rather than avoid drought through increased water-use efficiency [65]. While a short growth period is correlated with reduced yield potential, if the specified cultivar's target environment represents a definitive stress area, then the early genotype gain under stress outweighs its potential yield deficiency.

From the systematic review results, it is evident that our understanding of the molecular mechanisms underlying drought tolerance in *C. lanatus* spp. is limited to the accumulation of abscisic acid (ABA) and heat shock proteins (HSPs). The literature has not explained phytosterols' role (a group of hormones that are essential for regulating plant development and morphogenesis) under drought stress in citron watermelon. Future genome-scale studies involving stress signaling pathways in *C. lanatus* spp. are necessary, given that drought response is dependent on species and genotype. Data obtained from such experiments can be applied to build network models, establishing a link between phenotypic traits with regulatory mechanisms. In addition to this, current and future generation DNA sequencing technology, high-throughput phenotyping platforms, and improved informatics resources expediting gene discovery will be key for improving abiotic stress tolerance in this crop.

Citron watermelon, a C3 xerophyte, can survive absolute moisture stress [76–78]. Despite its potential utility as a source of genes for drought tolerance breeding, the functional genomics of *C. lanatus* spp. has been limited by the lack of genetic approaches and complex-

ity of the phenomenon, and there is a need to bridge this gap. However, with the realization of "omics" technologies, it is possible to provide a comprehensive description of changes in the transcript, proteome, and metabolome levels during drought stress. Combining the data on phenomics and genomics should lead to a systems biology approach and identify target genes and critical metabolic pathways. This process's complete elucidation would enable interpreting the incredible nature of C3 xerophytes.

### 4.1. Citron Watermelon Contribution to Human Nutrition and Health

Citron watermelon could potentially fulfill nutritional requirements by supplying biochemical compounds with health-promoting properties. In watermelon seeds (the same family with citron watermelon), four proteinogenic amino acids (phenylalanine, threonine, tryptophan, and valine) have been identified, and proteins are accumulated in quantities higher than those found in cereals (10–15%). Watermelon seeds also contain vitamins A, C, D, E, and K and several antioxidants such as flavonoids [79]. Populations consuming flavonoids-enriched foods have low cancer frequency [80]. In addition, lactating mothers eating citron watermelon seeds can produce higher quality milk, as seen in animal models fed with isoflavone-rich fodder [81]. Flavonoids can inhibit degenerative diseases such as coronary heart disease, atherosclerosis, cancer, diabetes, and Alzheimer's disease through their antioxidant activity and modulating multiple protein functions [82]. In addition, the absence of gluten in citron watermelon seeds offers alternative nourishment for the celiac population (people with a reaction to eating gluten), and it could counteract the increasing problems of obesity in the developed world. In less developed countries, citron watermelon could significantly reduce malnutrition and death by hunger.

### 4.2. The Preservation of Citron Watermelon Biodiversity

Citron watermelon seeds of different accessions are currently being conserved in several seed banks worldwide (ex situ conservation). However, preserving agrobiodiversity means preserving indigenous farmers' associated culture living in the Sub-Saharan Africa region [83]. The importance of seed banks in the conservation of biodiversity is well known, and the success of future conservation and breeding programs hinges on the preservation of this diversity on-farm. Moreover, the transfer of indigenous knowledge and associated practices will help adapt citron watermelon to new regions. Citron watermelon is a crop of family heritage; knowledge is acquired from the parents who have cultivated the crop since childhood [84]. Mujaju et al. [85] pointed out that farmers in Sub-Saharan Africa are key role players in the preservation of genetic diversity of citron watermelon in their fields, and they have the expertise for the agronomic management of their accessions.

Industrial development is causing migration from rural areas to the cities [86]. In addition to the increasing demand for daily calories, this social and economic situation changes land use and increases the crop's genetic homogeneity. Due to better profits from staple crop exports and higher incomes from commercial farmers, small farmers migrate, putting their cultural and agro-biodiversity heritage at risk [87]. Therefore, it is of primary importance to preserve small-scale farming, where the greatest genetic diversity of citron watermelon and associated human culture is found [88]. In Southern Africa, citron watermelon is still grown in the major historical areas of cultivation, "the province of Limpopo, the province of Matabeleland, the province of Manicaland and Omaheke region," which are an integral part of rural cultural heritage and identity [89,90]. Citron watermelon is a promising crop in a broader context. However, scientists and stakeholders must do all they can to preserve the heritage of citron watermelon so that this crop can continue to be cultivated while contributing to food quality and security in the Sub-Saharan Africa and globally.

## 5. Future Research Perspectives

Building on the literature examined, we identify six priority areas for research (summarized in Figure 4) and make recommendations for the short and long-term development

of citron watermelon as a crop that could contribute to food security and changing climatic conditions.

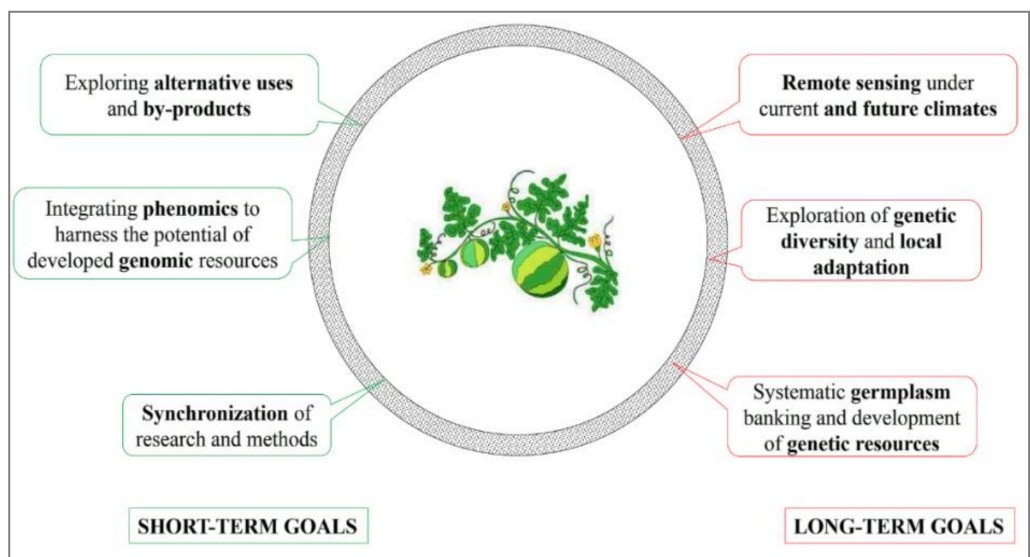

**Figure 4.** Roadmap for the sustainable development and exploitation of citron watermelon for food security and to support livelihoods (Author's drawing).

### 5.1. Synchronization of Research and Methods

According to the Global Biodiversity Information Facility (GBIF) (https://www.gbif.org/) (accessed on 8 February 2020) [91,92], citron watermelon research and the germplasm record include researchers from at least ten institutions in 13 countries. Despite positive national and international collaborations [93], citron watermelon research is still disconnected with interesting and relevant research programs running in isolation [4,94]. This review aims to draw together many disparate aspects of citron watermelon research to facilitate researchers' body knowledge and collaboration. In addition, we relate the experience of the BamNetwork (http://bambaragroundnut.org/ (accessed on 15 June 2020)) [95], the online representation of the international research community on Bambara groundnut (*Vigna subterranea* L.), which sought to bring together the expertise and enable close collaboration, the sharing of materials, resources, data, and technology. It is our view that citron watermelon research and food security in SSA could benefit from such an approach, with equitable and appropriate access and benefit-sharing agreements in place. Here, the suggestion is to develop a web database, which will act as an open repository for data emerging from citron watermelon research programs.

### 5.2. Integrating Phenomics to Harness the Potential of Developed Genomic Resources

The most productive farming land is facing biotic and abiotic stresses (fungal and bacterial diseases, heat, salinity, and drought stresses) [96]. All these biotic and abiotic stresses exert tremendous survival pressure on crops. Under the prevailing conditions and available resources, new plant varieties with desired traits (drought tolerance) and high yield potential need to be developed. This can be achieved by understanding the genetic makeup of plants (genomics) and their phenotype (phenomics) and the interaction between them in different environments.

In this era of phenomics, high-throughput precise phenotyping helps to amass high-quality, accurate phenotyping data. The high-quality phenotypic data are useful for meaningful genetic dissection and genomics-assisted breeding for drought stress tolerance. The earlier use of destructive plant phenotyping methods now gives way to high-throughput non-destructive, precise imaging techniques. Several phenomics platforms [97] are now available with facilities allowing scientists new windows into living plants' inner machin-

ery [98]. These facilities embrace (i) infrared cameras to scan temperature profiles and transpiration, (ii) incandescent microscopy to measure photosynthesis, (iii) 3D camera to record precise changes in growth responses after crop plants are exposed to stresses, (iv) lidars (light detection) to measure growth rates, and (v) magnetic resonance imaging (MRI) to examine root and leaf physiology.

### 5.3. Exploring Alternative Uses

Citron watermelon has the potential to produce other valuable by-products. The leaves can be cooked as vegetables, and seeds can be roasted as snacks, ground into a powder, and used as a condiment. Exploring alternative uses requires indigenous knowledge from the local farmers where the crop is grown through conducting ethnobotanical surveys. In India, high-value protein and oils are extracted from a watermelon known as Mateera Beej [99]. The seeds contain 35–50% crude protein, 28–40% oil, and minerals in significant quantities. Further, the oil contains more than 80% unsaturated fatty acids, with linoleic acid being the dominant fatty acid (68.3%). In citron watermelon, the chemical basis and nutrient composition have not been explored. Alka et al. [100] reported pharmacological activities (treatment of urinary tract infection, bed wetting, dropsy, and renal stones) of *Citrullus lanatus*.

### 5.4. Remote Sensing under Current and Future Climates

Estimates of the land area under citron watermelon cultivation [101] and associated yields are highly variable. They have been hampered by poor record-keeping and difficult access to remote areas. The short-term nature of citron watermelon cultivation, local differences in cultivated landraces, plant growth rates, agronomic practices, and dependency on co-staple crop productivity in any given period make estimating citron watermelon production difficult. Therefore, standardized empirical analyses for both land areas suitable for citron watermelon cultivation and the area currently under cultivation, yield components, and inter-annual trends are lacking.

Improvements in the resolution and accessibility of satellite data from National Aeronautics and Space Administration (NASA) products such as Moderate Resolution Imaging Spectroradiometer (MODIS) and Sentinel 2 are increasingly applied to vegetation and crop surveys [102,103]. Therefore, there may be a potential to use freely accessible satellite data to monitor citron watermelon production directly in the near term. Furthermore, this approach could be applied to mapping the crop suitability of citron watermelon. Concomitantly, upgraded regional bioclimatic datasets (Worldclim2) and an improved network of climate stations and data loggers will allow better characterization of the citron watermelon environmental niche and stress conditions. The impact of climate change under a range of future scenarios is yet to be quantified for citron watermelon and will form an essential part of any future development strategy.

### 5.5. Exploration of Genetic Diversity and Local Adaptation

Citron watermelon genetic diversity distributed across different environmental conditions indicates that the process of domestication might have facilitated the adaptation of landraces to local conditions, and indeed to a wider range of conditions than its wild progenitor. Since citron watermelon is propagated by seed, this represents a robust system to investigate the genomic basis of drought adaptive traits. Key steps to achieve this would be the characterization of existing citron watermelon genetic diversity using high-resolution genomic markers, standardized methods to measure fitness and yield, and robust environmental conditions monitoring. Concurrently, assessing the risk of erosion of citron watermelon genetic diversity through the loss or decline of landraces should be a priority for future citron watermelon monitoring strategies. In the medium term, this could similarly be extended to monitoring of crop wild relative diversity. In the long term, with the prerequisite knowledge of germination biology, novel sexual breeding using mapping populations and pan-genomic sequencing may enable the development of improved

genotypes that are tolerant of diseases, better adapted to current and future climates, and have desirable yield and by-product attributes.

*5.6. Systematic Germplasm Banking and Development of Genetic Resources*

*Citrullus lanatus* species, particularly citron watermelon landraces, are currently severely underrepresented [90]. This chronically reduces the potential for plant breeding and crop improvement. In the long term, under scenarios of habitat loss, agricultural intensification, disease spread, climate change, and the introduction of high-yielding genotypes, citron watermelon as an invaluable plant genetic resource is at risk of losing genetic diversity and consequently leading to the loss of genetic diversity.

While many landraces are present among subsistence farmers in Sub-Saharan Africa, citron watermelon germplasm management is vulnerable to outcrossing and poor documentation, and it needs commitment to proper maintenance [94]. Therefore, further exploration of germplasm banking's potential from a wide range of spatial and environmental conditions is a crucial research objective. Conventional breeding and ex situ seed conservation also require an understanding of desiccation, longevity of storage, and, essentially, the germination requirements. As with citron watermelon, much of this is not well understood. With appropriate access and benefit-sharing agreements, germplasm could be incorporated into established seed banks, benefiting research and sustainable exploitation and safeguarding an essential tropical crop.

In Nepal, a digital information system is currently under development as part of the Nepal Seed and Fertilizer (NSAF) project, which is funded by the United States Agency for International Development (USAID) [104]. This system allows easy access to an electronic seed catalogue with all registered varieties' features and sources; simultaneously, the balance sheet collects and shares information on seed demand and supply by all stakeholders in real time.

## 6. Conclusions

Under the context of climate change and crop production, citron watermelon is an interesting plant species whose capacity to tolerate adverse environmental conditions (water stress) and remarkable nutritional qualities warrant further research in all fields of plant biology, agronomy, and ecology. We projected short-term and long-term goals integrating the fundamental factors that explain and determine the future of citron watermelon regarding food security, biodiversity conservation, and crop diversification. Additionally, smallholder farmers should be encouraged to rely on a broader range of genotypes to sustain small-scale crop production and their economic, social, and cultural interactions. This will reinforce local conservation dynamics and ensure the sustainability of citron watermelon locally and around the world.

**Author Contributions:** T.M., A.O.O. and J.M. did the paper's initial conceptualization. T.M. then led the paper's write-up, and all authors then reviewed and approved the paper before publication. All authors have read and agreed to the published version of the manuscript.

**Funding:** This research received no external funding.

**Acknowledgments:** The authors acknowledge the University of KwaZulu-Natal Research Office for subscribing to Scopus for easy access to publications.

**Conflicts of Interest:** The authors declare no conflict of interest.

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
