# Peer review of "Citron Watermelon Potential to Improve Crop Diversification and Reduce Negative Impacts of Climate Change"

_sustainability, doi:10.3390/su13042269_

Round 1
Reviewer 1 Report
I read with interest your manuscript entitled “Citron watermelon potential to improve crop diversification and reduce negative impacts of climate change”.
I have some comments or suggestion in order to improve the manuscript.
Line 10: “Citrullus lanatus var. citroides”: I would suggest to insert the authority names (abbreviations) for the species name.
Line 11: “sub-Saharan” or “Sub-Saharan” (line 20)? Please standardize in the manuscript.
Line 23: Keywords should not be the repetitions of the title words.
Line 17: I would suggest to use the full species name.
Line 40: I would suggest to remove “(i)”, “(ii)” and “(iii)”.
Lines 46-47: Please rephrase the sentence.
Line 48: I would suggest to delete “(natural gelatin)”. The meaning of the term “pectins” is well known.
Lines 65-66: I would suggest to change “local/indigenous” to “local and indigenous”.
Lines 67-68: I would suggest to remove “(i)” and “(ii)”.
Line 68 : “do not provide resources (funds)…”: Please improve this sentence.
Line 74: I would suggest to avoid using “/”.
Lines 91-96: I would suggest to avoid writing: “firstly”, “secondly”, “thirdly”, and “fourth”.
Lines 103, 108: Please change to “Koutsos et al.”
Lines 104-105: I would suggest to use the numbering according to the Microsoft Word template (Instructions for Authors).
Lines 107, 156, 177, 290: Use dot after the figure number. Please change e.g. „Figure 1:” to „Figure 1.” (see the Microsoft Word template).
Line 109: I would suggest to number the subsections e.g. 2.1. Research question, 2.2. Data sources and search strategy etc. (see the Microsoft Word template).
Line 110: Please change semicolon to colon and remove left double quotation mark or rephrase the sentence.
Lines 103-105, 115-117, 184-187, 306-310: I would suggest to use bulled or numbered lists (see the Microsoft Word template).
Lines 123, 166: Use dot after the number. Please change e.g. „Table 1:” to „Table 1.” (see the Microsoft Word template).
Lines 124-125: Remove or add quotation marks. Standardize.
Line 148: I would suggest to number the subsections.
160: “C. lanatus”: I would suggest to use the full scientific name.
Line 166: I would suggest to change the title using the following terms: drought avoidance (DA), drought tolerance (DT), and drought escape (DE), with the abbreviations in parentheses.
Lines 193-194: I would suggest to add the authority (e.g. “L.”) for the species name: Alyssum alyssoides (L.).
Line 213: “Citron watermelon a C3 xerophyte”: Please correct.
Line 224: I suggest to improve the subsection title.
Line 237: “the celiac population (people with reaction to eating gluten)”: please rephrase the sentence.
Line 245: I would suggest: the Sub-Saharan African region (SSA) or Sub-Saharan Africa (SSA). The name should be used when first mentioning in a section or a subsection.
Lines 248-250: Missing dot. Please rephrase the sentence.
Line 250: Please change „Mujaju, Zborowska” to „Mujaju et al.”
Lines 251-253: I would suggest to remove “(i)” and “(ii)”.
Lines 256-257: “…(few genotypes are grown for commercial purposes)”: I would suggest to rephrase the sentence.
Lines 262-263: I would suggest to remove the parentheses and to add colon (e.g. “such as province of Limpopo, the province of Matabeleland and the region of Omaheke”).
Line 282: “Vigna subterranea”: I would suggest to add the authority (e.g. “L.”) for the species name.
Line 293: I would suggest to add “and” instead of “/”.
Line 305: “https://www.plantphenomics.org.au/” should be included in the reference list. In the text, the reference number in brackets should be placed instead of https://www.plantphenomics.org.au/.
Line 310: “root/leaf”: I would suggest to add “and” instead of “/”.
Line 317: “Mateera Beej”: Please specify (watermelon, popularly known as Mateera Beej).
Line 320: Please change „Alka, Anamika” to „Alka et al.”
Line 322: “C lanatus”: I would suggest to use the full scientific name.
Line 325: I would suggest to remove “(i)” and “(ii)”.
Lines 401-581: The references list is not formatted according to journal criteria. Please check all citations in the text and references carefully.
Author Response
I have attached a document with responses to reviewer's comments

Reviewer 2 Report
My calculations show that the period under consideration 1995-2019 covers 25 years, and not - as the authors say (line 152) - 24 years. Consequently, on line 157, instead of "In the past 24 years ..." it should read "In the past 25 years ...".
Lines 151-152 read "Table 2 summarize 61 studies from a systematic review on drought adaptation mechanisms ...". This information is not clear for me because Table 2 lists 52 publications. If study and publication mean the same thing, then something is wrong here. The list of publications and their summary in the last column in table 2 shows that some publications cover more than one area of consideration, i.e. two or even three (article [31]) of DA: drought avoidance DT: drought tolerance DE: drought escape. Therefore, in the description and discussion of data in Table 2, it is necessary to present in detail what resulted in the number of 61 studies. Summing up the markings from the last column of Table 2, we actually get 61. It would be worth mentioning that these are 61 studies presented in 52 articles (publications). As a result, the information presented will be clearer. On the other hand, however, I have some doubts about the number of 61 studies, because in Table 2 in the case of one of the publications, i.e. "Identifying sources of water stress tolerance from wild species of the family Cu-curbitaceae in vitro culture" [48], it was not assigned none of the areas within DA: drought avoidance DT: drought tolerance DE: drought escape. In my opinion, this requires correction or clarification.
Lines 151-152 read "Table 2 summarize 61 studies from a systematic review on drought adaptation mechanisms ...". Meanwhile, the title of Table 2 is "Summary of studies on drought adaptation strategies ...". Perhaps it would be worthwhile to unify the phrases used and to write the same word in both indicated places, ie "mechanisms" or "strategies".
Were Figures 3 and 4 made by the authors of the article? Perhaps it would be worth writing it in the captions to the drawings.
In my opinion, in the discussion part it would be worth mentioning how the production of Citron watermelon fits into the plant rotation system. This is particularly important from the point of view of plant production planning by farmers, mentioned in the article. Is the Citron watermelon grown in the same field over the years or is it recommended to alternate the cultivation of the Citron watermelon with other plants. This is important information for farmers. What condition of the soil does the Citron watermelon leave behind and what soil (after which plants) does it require? These aspects should be briefly elaborated on in the article due to their practical nature, important for farmers.
Author Response
I have uploaded a Word document with responses to the reviewer's comments

Reviewer 3 Report
Please see attachment.

Author Response

(The authors gave the same response as above.)

Round 2
Reviewer 3 Report
I am OK with the edited version.